# Vascular Aspects in Glaucoma: From Pathogenesis to Therapeutic Approaches

**DOI:** 10.3390/ijms22094662

**Published:** 2021-04-28

**Authors:** Anna-Sophie Mursch-Edlmayr, Matthias Bolz, Clemens Strohmaier

**Affiliations:** Department of Ophthalmology/Optometry, Johannes Kepler University, 4020 Linz, Austria; anna-sophie.mursch-edlmayr@kepleruniklinikum.at (A.-S.M.-E.); matthias.bolz@kepleruniklinikum.at (M.B.)

**Keywords:** glaucoma, blood flow, intraocular pressure

## Abstract

Glaucomatous optic neuropathies have been regarded as diseases caused by high intraocular pressure for a long time, despite the concept of vascular glaucoma dating back to von Graefe in 1854. Since then, a tremendous amount of knowledge about the ocular vasculature has been gained; cohort studies have established new vascular risk factors for glaucoma as well as identifying protective measures acting on blood vessels. The knowledge about the physiology and pathophysiology of the choroidal, retinal, as well as ciliary and episcleral circulation has also advanced. Only recently have novel drugs based on that knowledge been approved for clinical use, with more to follow. This review provides an overview of the current vascular concepts in glaucoma, ranging from novel pathogenesis insights to promising therapeutic approaches, covering the supply of the optic nerve head as well as the aqueous humor production and drainage system.

## 1. Introduction

Glaucoma (or glaucomatous optic neuropathy, GON) refers to a group of diseases characterized by retinal ganglion cell atrophy with corresponding visual field defects. If unrecognized or untreated, glaucoma can lead to severe visual impairment. It is amongst the most frequent causes of blindness in industrialized countries [1]. 

Elevated intraocular pressure (IOP) is the most important risk factor for glaucoma, and lowering it is the only proven treatment to date [2,3,4,5]. Besides elevated IOP, vascular abnormalities have been recognized as a possible cause for GON, with the concept of vascular glaucoma dating back almost 170 years [6]. Those 170 years have brought along many advances in the understanding of the vascular aspect of glaucoma: cohort studies have established new vascular risk factors as well as identifying protective measures acting on blood vessels. New technologies enable a more refined investigation of the ocular vascular beds, steadily increasing our understanding of the physiology and pathophysiology. Only recently have novel drugs based on that knowledge been approved for clinical use, with more to follow.

This review aims to provide an overview of selected vascular aspects in glaucoma, covering vascular risk factors in epidemiological studies as well as physiological aspects in animal models and abnormal vascular findings in glaucoma patients. 

## 2. Vascular Risk Factors from Cohort Studies 

Several cohort studies conducted in different populations have identified vascular risk factors for various aspects of glaucoma, but findings are inconsistent.

Low ocular perfusion pressure (OPP, either mean OPP or diastolic OPP, or both) has been identified as a risk factor for OAG in the Los Angeles Latino Eye Study [7], the Egna-Neumarkt Study [8], the Singapore Eye Study [9], and the Barbados Eye Study [2,10]. The Early Manifest Glaucoma Trial identified low systolic perfusion pressure as a risk factor for disease progression independent of IOP [4]. Surprisingly, other studies did not confirm an association of low systolic perfusion pressure with NTG or OAG with low IOP [8].

A recent meta-analysis found a consistent association of low perfusion pressure and POAG but not NTG [11], even though some large epidemiologic studies were excluded due to mode of data reporting. The Leuven Eye Study found higher mean OPP values in glaucoma patients in a cross-sectional design [12]. 

An important caveat is the heterogeneity of the blood pressure (BP)-related formulas used in different studies. A report by Barbosa-Breda et al. found differences of up to 100% for the obtained OPP values depending on the applied formula [13]. Furthermore, ocular perfusion pressure is commonly calculated using arterial blood pressure and IOP (see the next section for a more detailed discussion), and its dependence on IOP was reported with inconsistent findings [7,14].

## 3. Ocular Blood Supply and Ocular Perfusion Pressure

The blood supply of the ocular tissues arises from the ophthalmic artery, the first major branch of the internal carotid artery, as it supplies the human orbit. The ocular branches of the ophthalmic artery can be divided into the central retinal artery and the posterior and anterior ciliary arteries [15]. The central retinal artery supplies the retina as well as the superficial layers of the optic nerve head (ONH). The posterior ciliary arteries divide in the long (LPCAs) and short posterior arteries (SPCAs). The former supply the iris, the ciliary body, and the anterior choroidal tissue, whereas the latter feed the posterior choroid and the largest part of the anterior ONH [15,16]. The anterior ciliary arteries are responsible for supplying blood flow to the anterior uvea. Blood supply of the ONH is complex as it can be stratified into four layers (Figure 1). The superficial nerve fiber layer consists mainly of the axons of the retinal ganglion cells and is supplied by the inner retinal circulation. The prelaminar region of ONH receives its blood supply from the branches of the peripapillary choroid and from the SPCAs. The third layer is the lamina cribrosa (LC), which is exclusively supplied by the SPCAs. The retrolaminar region located outside the globe is nourished by both the pial vessels and the SPCAs [17].

As in any tissue of the body, arterial blood pressure drives blood through the vasculature, and the combined resistance of all vessels causes a pressure drop at the venous side of the circulation (i.e., the vortex veins or the orbital veins in the case of episcleral vessels). A major difference to most circulations, however, is the compressing force exerted by intraocular pressure (IOP). This causes the vortex veins to behave as Starling resistors, i.e., maintaining an intraluminal pressure slightly higher than IOP to prevent them from collapsing. Figure 2 illustrates the effect of intraocular pressure (as an external compression force) on the flow and transmural pressure in an intraocular vessel [18]. 

Assuming that the venous pressure must be slightly higher than the intraocular pressure, the ocular perfusion pressure (i.e., the pressure gradient driving blood flow) is commonly approximated by the following formula [19]:OPP = MAP − IOP
(MAP = mean arterial pressure).

This approximation, however, might not be accurate in clinically relevant pressure ranges. There is some evidence suggesting that the Starling resistor behavior outlined above may produce non-linear results, especially at low IOP values. Maepea was the first to note that choroidal venous pressure appears to be significantly higher than IOP at low IOP values [19].

Currently, human in vivo measurements of retinal or choroidal venous pressure are technically not possible, despite approximations of retinal venous pressure existing with ophthalmodynamometry [20]. Unpublished data obtained in a rabbit model, kindly provided by Reitsamer, on the relationship between IOP and choroidal venous pressure showed that with IOP values in a normal range, choroidal venous pressure clearly deviates from the assumed 1:1 relationship; thus, perfusion pressure in the choroid is less than estimated by the PP = MAP − IOP formula (Figure 3). From a clinical perspective, this is of high relevance in normal tension glaucoma, where low (diastolic) blood pressure might cause insufficient blood supply of the optic nerve head. Furthermore, this might cause an overestimation of perfusion pressure in clinical studies.

## 4. Regulation of Ocular Blood Flow

The arteries of the choroid, retina, ciliary body, and optic nerve head circulations are all capable of active vascular resistance regulation, commonly termed autoregulation [21,22,23,24,25]. This includes adaptions to metabolic changes (metabolic autoregulation) and adaptions to changes in transmural pressure (myogenic autoregulation), as well as neuronal autoregulation, a somewhat broader term including both paracrine and neurohumoral regulatory mechanisms [18,21,26,27,28]. The orbital veins, on the other hand, exhibit a passive behavior during perfusion pressure changes [29]. A striking clinical example is the immediate choroidal thickness increase during the Valsalva maneuver [30], resulting in increased intraocular pressure. Passive venous behavior also implies an effect of elevated thoracic pressure, as it occurs in obstructive sleep apnea syndrome as well as in obesity, on the ocular circulation. Orbital veins are difficult to measure in most species with the exception of rabbits, where a foramen in the scull allows direct cannulation [29]. Interestingly, in this model, a linear relationship between orbital venous pressure and mean arterial blood pressure was found.

## 5. Episcleral Circulation and Episcleral Venous Pressure

The episcleral circulation is of high relevance for glaucoma because of its impact on intraocular pressure. Steady-state intraocular pressure (IOP) can be described by the Goldmann equation as the relationship between aqueous flow, uveoscleral outflow, outflow facility, and episcleral venous pressure (EVP) as an additive factor [31]. EVP is the pressure that has to be overcome by fluid to leave the eye via the conventional (trabecular) outflow pathway and, therefore, is an important determinant of steady-state IOP. The anatomical organization of the episcleral vessels appears to be well suited for regulation of the pressure in the episcleral veins and thereby IOP, as the episcleral circulation is mostly devoid of capillaries and arteries, and veins are connected through anastomoses [32,33]. More interestingly, these anastomoses are supplied by nerves, whose nerve endings stain positively for autonomic neurotransmitters [34]. A study by Zamora and Kiel confirmed Ascher’s original observation that the episcleral circulation responds to topical anesthetics in a rabbit model [35]; however, this could not be replicated in humans [36]. Additional data supporting the assumption that the EVP is not passive, but actively regulated, came from a study investigating EVP changes during changes in body posture, where EVP changed less than predicted by the hydrostatic water column effect [37]. To date, the functional significance of the innervation of the episcleral circulation is only scarcely investigated, most likely due to technical reasons. However, one study reported changes in EVP in response to stimulation of a brain stem nucleus [38], thus indicating the possibility for active regulation.

The therapeutic potential of the episcleral circulation was recently confirmed in an experimental study with cromakalim prodrug 1 (CKLP1) in a mouse model. CKLP1 reduces EVP and thereby EVP–as predicted by the Goldmann equation. This effect is additive to other known substances acting on aqueous humor formation [39].

## 6. Measurement of Ocular Blood Flow

Non-invasive measurement of ocular perfusion has been the focus of intense research interest in recent decades. A wide range of techniques all utilizing the Doppler effect have been introduced (Table 1). The Doppler effect describes the phenomenon of change in frequency and wavelength of a wave caused by the change in distance between the creator and observer of the wave. For light, this causes a shift in color. The faster the creator is moving towards the observer, the greater the blue shift, whereas sound becomes higher in pitch. In vascularized tissue, the Doppler effect is caused by moving red blood cells within the vessels [40]. With adequate technical equipment, the change in frequency or wavelength can be detected and computed into parameters describing perfusion. Absolute measurements of perfusion are only obtained if information on the vessel diameter is provided, taking into account the Hagen–Poiseuille equation. 

Ultrasound-based color Doppler imaging is the only technology that measures the Doppler shifts in sound waves, whereas other techniques such as laser Doppler flowmetry, laser Doppler velocimetry, and Doppler optical coherence tomography use laser light (see Table 1).

Some of the techniques to measure perfusion of the ONH are limited to superficial tissue, supplied by the central retinal artery, but results from an animal model suggest that not only the superficial but also the deep ONH vascular supply is associated with pathologic changes in glaucoma [41]. Contributions from deeper ONH layers to the signal vary depending on the techniques, as different wavelength light sources are used [42] (see Table 1).

The laser speckle flowgraphy (LSFG) method is based on the laser speckle phenomenon. It describes an interference phenomenon occurring when a diffusing surface (e.g., the ocular fundus) is illuminated by a coherent light source (i.e., a laser). The backscattered light has the appearance of a granular pattern (i.e., the speckle pattern), which is entirely random and can only be described statistically [43]. In the case of moving scatters (e.g., red blood cells) within the illuminated field of view, the appearance of the speckle pattern changes rapidly; the greater the blood velocity, the higher the rate of variation in the speckle pattern in the vascular area and the lower the speckle contrast [44]. This enables two-dimensional measurements of perfusion at the ONH, the retina, and the choroid.
ijms-22-04662-t001_Table 1Table 1Overview of different technologies to measure ocular perfusion.TechnologyPrincipleOutcome ParameterMeasurement AreaAdvantagesLimitationsPenetration Depth (in Humans)Laser Doppler Flowmetry (LDF)Laser (780 nm) Mean VelocityVolumeFlow100 × 100–400 × 400 µmCommercially availableValid reproducibility [45]No absolute measurements300–400 µm [46]Laser Doppler Velocimetry (LDV)Laser (675 nm)Maximal Velocity Individual retinal arterioles/venulesAbsolute measurementsCommercially available Prone to eye movementsComplex and time-consuming measurementsNo OHN measurements0.5–0.8 mm [47]Color Doppler Imaging (CDI)Ultrasound (Doppler with 6.5 MHz) Peak systolic velocity (PSV)End diastolic velocity (EDV)Mean flow velocity (MFV)Resistivity index (RI)Retrobulbar arteries (ophthalmic artery, retinal central artery, short posterior ciliary artery) Commercially availableTime-consumingLimited to retrobulbar arteriesNo absolute measurementsNo measurement standardPatient in supine position during measurementRetrobulbarDoppler OCT (D-OCT)Laser (841 nm)Total volumetric RBFRetinal vesselsAbsolute measurementProne to artefacts due to eye movementNot commercially available1–2 mm [47]Laser Speckle Flowgraphy (LSFG)Laser (808 nm)Mean blur rateManually set area of interest (ONH, retinal vessels or choroid), or MBR for each pixelPatient and investigator friendlyCommercially availableNo absolute measurements Up to retrolaminar region [45]


## 7. Hemodynamic Alterations in Glaucoma Patients

With color Doppler imaging, reduced blood flow velocities in retrobulbar vessels were shown in both high- and low-tension POAG patients compared with healthy control subjects [12,48,49]. The use of laser Doppler velocimetry revealed that blood flow in the peripapillary capillaries, nourishing the superficial nerve fiber layer of the ONH, was reduced in patients with OAG and NTG when compared with a healthy group [50]. Results from scanning laser Doppler flowmetry (LDF) measurements in the disc cup and the neuroretinal rim showed significantly reduced perfusion in patients with high tension POAG compared with healthy subjects, as well as subjects with ocular hypertension [51,52,53]. Further, results obtained with LDF suggested a correlation of glaucomatous damage with reduced ONH perfusion [54].

Much attention has been given to the technology of OCT angiography [55]. However, it should be noted that this technology is unable to measure flow or perfusion but rather gives information about vessel density. 

Studies reporting results from LSFG measurements in eyes with normal tension POAG have been published for over a decade. Significant differences in ONH perfusion have been shown between patients with normal tension glaucoma and healthy individuals [56,57].

All reported studies above were conducted as cross-sectional trials. Results of a prospective, longitudinal study on structure rather than perfusion, comparing healthy individuals with patients with preperimetric glaucoma and glaucoma patients with visual field defects, have indicated that macular vessel density measured with OCT angiography declined faster than the ganglion cell thickness [58]. Longitudinal data regarding ONH perfusion in glaucoma came from a primate glaucoma model. Cull et al. performed a study to evaluate longitudinal changes in ONH blood flow with LSFG. They showed that basal ONH BF was strongly associated with the stage of glaucoma severity (measured by loss of RNFLT) and report a two-phase pattern of change. During the earliest stage of glaucoma, ONH blood flow exhibited a mild increase, after which it progressively declined with an increasing degree of glaucoma severity. When more than 40% of the RNFLT was lost, ONH blood flow was reduced by 25% below baseline [59]. The initial increase in ONH blood flow was confirmed by results from a cross-sectional study in humans [60]. 

Until recently, only one longitudinal study reported changes in optic nerve head perfusion over time in humans. In 2005, Martínez et al. published results from a longitudinal trial where they observed 49 patients with high tension POAG over 36 months. In total, 23 eyes had progressed during the study period. The authors identified increased resistivity in the ophthalmic artery and the short posterior ciliary arteries as risk factors for the progression of glaucomatous visual field defects [61]. In 2020, a retrospective, longitudinal study was published, which included 350 eyes of 225 POAG patients. The authors found older age, high pulse rate, and whether the damaged quadrant was superior or temporal to be risk factors for faster perfusion parameter deterioration. When two or more risk factors were obtained, MT decrease preceded RNFLT decrease [62]. This finding indicates that in a subgroup of glaucoma patients, reduced ONH perfusion is part of the pathogenesis. 

## 8. Novel Therapeutic Approaches for the Vascular Aspects of Glaucoma

Currently, there is no established therapy specifically addressing the choroidal or optic nerve head vasculature. Some IOP-lowering agents at least partly act through vasoconstriction in the ciliary body vasculature [63]. One interesting observation, however, hints towards a therapeutic potential for drugs acting on the ocular vasculature: increased oral uptake of dietary NO donors reduced the incidence of parafoveal visual field defects in a large cohort study [64].

Furthermore, the episcleral circulation recently emerged as a therapeutic target of new classes of IOP-lowering drugs as well as new routes of administration of existing drug classes.

Rho-kinase inhibitors are a novel drug class for glaucoma treatment for which FDA approval was received in 2017 [65]. They act at least partly through increasing outflow facility in the conventional (trabecular) pathway [66,67]. Additionally, an EVP-lowering effect has been shown for some Rho-kinase inhibitors in an animal model [68]. The clinical significance of the EVP-lowering effect is not entirely clear, but a combined mechanism is likely, as the magnitude of IOP decrease cannot be explained by either mechanism alone in humans [69]. 

Another well-established drug class has an effect on the ocular vasculature. Prostaglandin analogs (PGAs) are an established drug class for glaucoma treatment [70], increasing uveoscleral outflow to lower IOP. Recent studies, however, revealed that PGAs show a different dose–response relationship depending on their route of administration (reviewed in [71]). Current data suggest this is due to the opposing effects of prostaglandin analogs on the uveoscleral outflow pathway as well as the episcleral venous pressure [72,73]. While the IOP-lowering effect of topical prostaglandins appears to be limited by the opposing effect of increasing EVP, intracameral prostaglandins seem to dodge this limitation [74]. It is important to note that these observations are based on animal models and need further verification in clinical trials.

In summary, both defining and treating possible vascular glaucoma triggers remain challenging in clinical routine as well as from a scientific point of view. Emerging technologies to measure ocular perfusion (e.g., Doppler OCT or LSFG) in vivo have the potential to further elucidate the role of the ocular perfusion in the pathophysiology of glaucoma. Furthermore, current drug development utilizes obtained knowledge on the ocular vasculature and will likely be available in clinical practice soon.

## Figures and Tables

**Figure 1 ijms-22-04662-f001:**
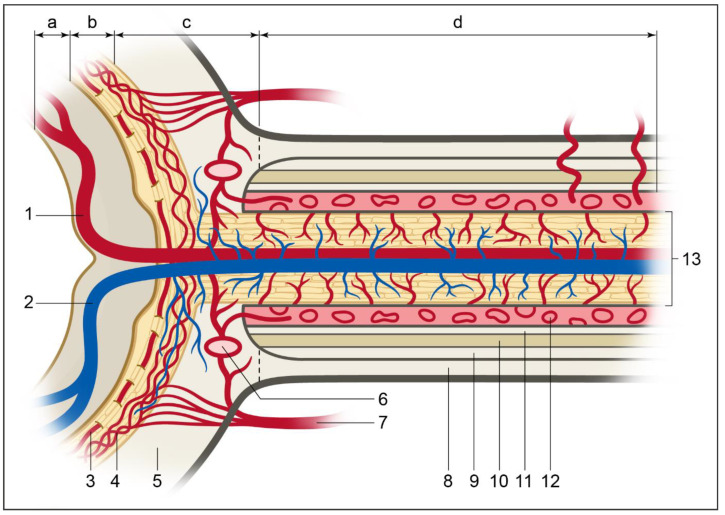
(a) Different areas of perfusion of the optic nerve head showing a superficial nerve fiber layer, (b) prelaminar region, (c) laminar region, (d) retrolaminar region, (1) central retinal artery, (2) central retinal vein, (3) retina, (4) choroid, (5) sclera, (6) circle of Zinn-Haller, (7) short posterior ciliary arteries, (8) optic nerve sheath (9) subdural cavity, (10) arachnoid mater, (11) subarachnoidal space, (12) pia mater, (13) optic nerve.

**Figure 2 ijms-22-04662-f002:**
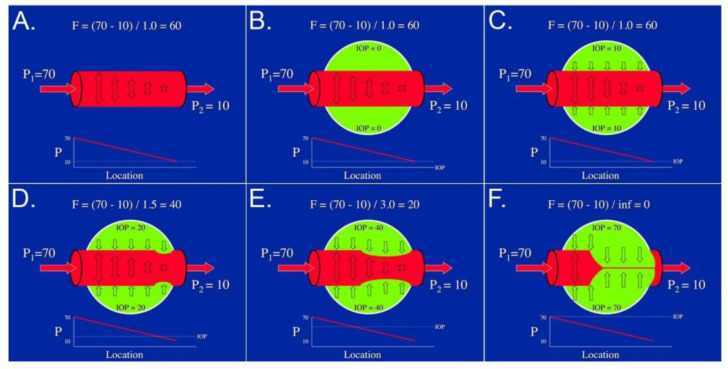
Ocular Starling resistor. (**A**) Vessel flow (F) is a function of the pressure gradient (P1—P2) along the vessel divided by the resistance. (**B**,**C**) If the vessel passes through an organ (e.g., the eye) with a low tissue pressure (e.g., IOP), the pressure inside the vessel exceeds the pressure outside the vessel (i.e., the transmural pressure gradient) and so the vessel remains distended. (**D**,**E**) If the tissue pressure is somewhat higher and exceeds the pressure at the lowest point inside the vessel (i.e., at the “venous” end), that region of the vessel will begin to collapse. This will increase the resistance to flow in that segment, thereby raising the intraluminal pressure until the transmural pressure becomes slightly positive again. (**F**) If the tissue pressure becomes greater than the arterial input pressure, the vessel will collapse completely, the resistance will be infinite, and flow through the vessel will cease. Reproduced with permission from [18].

**Figure 3 ijms-22-04662-f003:**
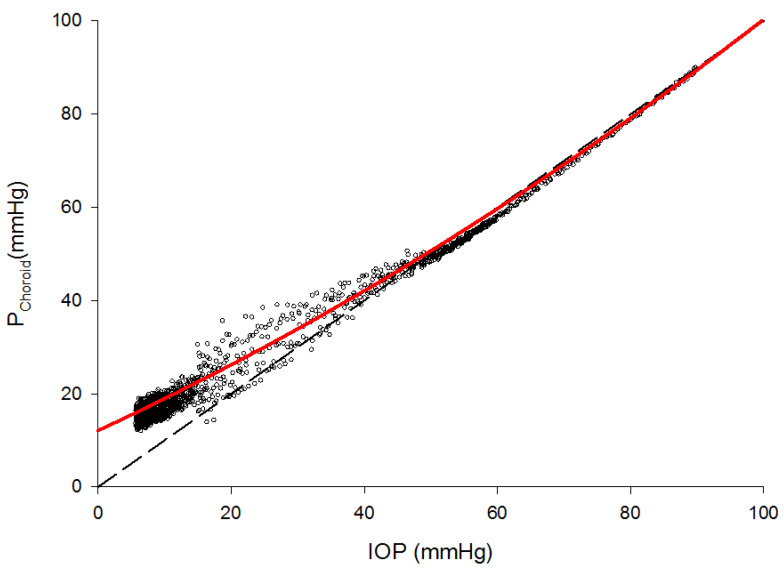
Relationship between intraocular pressure (IOP) and choroidal venous pressure (P choroid). As expected by the Starling resistor effect, choroidal venous pressure slightly exceeds intraocular pressure at medium to high IOP values. At lower IOP values, however, choroidal venous pressure deviates from this 1:1 relationship significantly, reaching 50% at values below 10 mmHg (unpublished observation by Reitsamer). Each dot represents a single pressure measurement, the red line represents a curve-fit of all measurement values.

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
