# Peer review of "Vascular Aspects in Glaucoma: From Pathogenesis to Therapeutic Approaches"

_ijms, 2021, doi:10.3390/ijms22094662_

Round 1
Reviewer 1 Report
The topic of the article sounds interesting but in my opinion there are some concept that need to be implemented by the authors.
Page 4 line 108: Authors referred to unpublished data, are these informations presented in a poster or in a book? I think it would be better to cite only published paper and verified by a correct process to avoid misleading message.
In my opinion section 4, 5 and 6 should be implemented becouse in my opinion represent the key part of this review and i find them to summarized.
There are some english errors that need to be addressed:
page 2 line 54 'the formula applied' I believed is the applied formula.
page 3 line 86-87 this sentence is not clear
page 4 line103 'Maepea was first to note' should be change in Maepea was the first to note.
Page 2 and 3 there are errors in references line 69 and 86 respectively.
Author Response
We’d like to thank both reviewers for their time and effort to improve our manuscript. All changes in the manuscript - except for added references- are highlighted by the “track changes” feature in Microsoft Word. Furthermore, all comments are addressed in a point-to-point fashion below. We hope that both reviewers find the revised manuscript appropriate for publication.
Reviewer 1
The topic of the article sounds interesting but in my opinion there are some concept that need to be implemented by the authors.
Page 4 line 108: Authors referred to unpublished data, are these informations presented in a poster or in a book? I think it would be better to cite only published paper and verified by a correct process to avoid misleading message.
This data will be published in a book, which is currently in the final steps of the editing process. We generally agree with the reviewer, that citations should be limited to published data. However, in this particular case we found the information to be of interest to the reader, as it provides a deeper insight into possible causes of the heterogeneity found in the literature. Furthermore, the figure rather refines previously published knowledge (Maepea et. al.).
In my opinion section 4, 5 and 6 should be implemented becouse in my opinion represent the key part of this review and i find them to summarized.
We agree with the reviewer that these sections are rather short, but there are reasons for it. Section 4 deals with the regulation of ocular blood flow. This topic is covered through numerous specialized review papers, of which some were cited already. However, to further guide the reader towards more focused review papers, we have added review articles from experts in the field. The topic of section 5 is covered in sufficient detail in our opinion, as section 8 extends the discussion of EVP and glaucoma therapy. Section 6 is also enhanced by the following table summarizing many aspects of ocular perfusion measurement.
There are some english errors that need to be addressed:
page 2 line 54 'the formula applied' I believed is the applied formula.
Has been corrected
page 3 line 86-87 this sentence is not clear
The sentence has been rephrased to make it clearer..
page 4 line103 'Maepea was first to note' should be change in Maepea was the first to note.
Has been changed.
Page 2 and 3 there are errors in references line 69 and 86 respectively.
We sincerely apologize for our sloppiness here, the error occurred while transferring our manuscript to the journal template.
Reviewer 2 Report
The review manuscript “Vascular aspects in glaucoma: from pathogenesis to therapeutic approaches” by Mursch-Edlmayr and co-authors describes the vascular aspects/risk factors in glaucoma and discuss recent studies on this matter. This review is well written, but some improvements could be made.
In general, all abbreviations, e.g. MAP, need to be introduced. Please revise the manuscript accordingly.
For the first part of section 3. some references should be added.
Figure 1: did the authors generate this figure?
Figure 1 and 2 need to be mentioned in the text.
Figure 3: add some information about the meaning of the red line as well as of the dashed line to the legend. The dots – single measurements? – also need to be explained in the legend.
Table: some parts of the table are hard to read. The width of the columns needs to be adjusted. The table might be easier to read without the dots for the bullet points.
An outlook and a short conclusion should be added at the end of the manuscript.
There seems to be an error with the citation system, e.g. line 69 and 86. The authors need to add the appropriate references and fix this.
Author Response
Reviewer 2
The review manuscript “Vascular aspects in glaucoma: from pathogenesis to therapeutic approaches” by Mursch-Edlmayr and co-authors describes the vascular aspects/risk factors in glaucoma and discuss recent studies on this matter. This review is well written, but some improvements could be made.
In general, all abbreviations, e.g. MAP, need to be introduced. Please revise the manuscript accordingly.
This has been changed.
For the first part of section 3. some references should be added.
We have added some references.
Figure 1: did the authors generate this figure?
Yes, this figure has been designed by the authors. Aron Cserveny’s help with the illustration is mentioned in the acknowledgement section.
Figure 1 and 2 need to be mentioned in the text.
This has been corrected – the refences where lost when the manuscript was transferred to the journal template.
Figure 3: add some information about the meaning of the red line as well as of the dashed line to the legend. The dots – single measurements? – also need to be explained in the legend.
The figure legend has been revised.
Table: some parts of the table are hard to read. The width of the columns needs to be adjusted. The table might be easier to read without the dots for the bullet points.
The table has been revised slightly to enhance readability.
An outlook and a short conclusion should be added at the end of the manuscript.
This has been added.
There seems to be an error with the citation system, e.g. line 69 and 86. The authors need to add the appropriate references and fix this.
We sincerely apologize for our sloppiness here, the error occurred while transferring our manuscript to the journal template.
Round 2
Reviewer 2 Report
The authors made all the necessary revisions. Nice review manuscript!
Author Response
We'd like to thank the reviewer for the time and effort to improve our manuscript